# Quantitative Evaluation of Apparent Diffusion Coefficient Values, ISUP Grades and Prostate-Specific Antigen Density Values of Potentially Malignant PI-RADS Lesions

**DOI:** 10.3390/cancers15215183

**Published:** 2023-10-28

**Authors:** Nadine Spadarotto, Anja Sauck, Nicolin Hainc, Isabelle Keller, Hubert John, Joachim Hohmann

**Affiliations:** 1Institute of Radiology and Nuclear Medicine, Cantonal Hospital Winterthur, 8401 Winterthur, Switzerland; nadine.spadarotto@ksw.ch; 2Clinic of Urology, Cantonal Hospital Winterthur, 8401 Winterthur, Switzerland; anja.sauck@ksw.ch (A.S.); isabelle.keller@ksw.ch (I.K.); hubert.john@ksw.ch (H.J.); 3Department of Neuroradiology, Clinical Neuroscience Center, University Hospital Zurich, 8091 Zurich, Switzerland; nicolin.hainc@usz.ch; 4Medical Faculty, University of Zurich, 8032 Zurich, Switzerland; 5Medical Faculty, University of Basel, 4056 Basel, Switzerland

**Keywords:** prostate cancer, diffusion-weighted MRI, apparent diffusion coefficient (ADC), ISUP grade, prostate-specific antigen density (PSAD)

## Abstract

**Simple Summary:**

This study demonstrated a correlation between the apparent diffusion coefficient (ADC) of diffusion-weighted images and potentially malignant prostate lesions, with even better correlation when using the ratio of ADC and prostate-specific antigen density (PSAD), or ADC/PSAD, as a biomarker. Threshold values were determined in order to distinguish between histological cancer grades. Retrospectively, 403 patients with a total of 468 prostate lesions were enrolled. All patients had undergone multiparametric magnetic resonance imaging (mpMRI) and a subsequent biopsy of their prostate lesions, whereby their histological cancer grade was determined. Lower ADC values and lower ADC/PSAD ratios correlated with higher histological cancer stages. Therefore, it was possible to distinguish between histological cancer stages, and thus to reduce the number of unnecessary biopsies.

**Abstract:**

The aim of this study was to demonstrate the correlation between ADC values and the ADC/PSAD ratio for potentially malignant prostate lesions classified into ISUP grades and to determine threshold values to differentiate benign lesions (noPCa), clinically insignificant (nsPCa) and clinically significant prostate cancer (csPCa). We enrolled a total of 403 patients with 468 prostate lesions, of which 46 patients with 50 lesions were excluded for different reasons. Therefore, 357 patients with a total of 418 prostate lesions remained for the final evaluation. For all lesions, ADC values were measured; they demonstrated a negative correlation with ISUP grades (*p* < 0.001), with a significant difference between csPCa and a combined group of nsPCa and noPCa (ns-noPCa, *p* < 0.001). The same was true for the ADC/PSAD ratio, but only the ADC/PSAD ratio proved to be a significant discriminator between nsPCa and noPCa (*p* = 0.0051). Using the calculated threshold values, up to 31.6% of biopsies could have been avoided. Furthermore, the ADC/PSAD ratio, with the ability to distinguish between nsPCa and noPCa, offers possible active surveillance without prior biopsy.

## 1. Introduction

Despite the established triage function of multiparametric MRI (mpMRI), the increased number of prostate mpMRIs being performed has resulted in an increased prostate biopsy rate [1,2,3,4,5]. This carries with it a complication rate of approximately 1%, a risk of false negative histology of up to 40% and possible underestimation of the degree of pathological differentiation [6,7,8,9]. In up to 77.9% of the targeted biopsied PI-RADS 4 lesions, no clinically significant prostate cancer (csPCa) is detectable, and in up to 60.9%, no prostate cancer (noPCa) is detectable at all. In PI-RADS 5 lesions, this is decreased to 27.6% and 13.1%, respectively. Thus, additional triaging biomarkers could prevent unnecessary biopsies [9,10].

Various studies dealing with the *quantitative* evaluation of mpMRI have shown a correlation of the apparent diffusion coefficient (ADC) value with the Gleason score, the PSA level and the clinical outcome [9,11,12,13,14,15,16]. The *qualitative* ADC evaluation included in the PI-RADS classification, however, is subjective and prone to high inter-reader variability, with agreement of up to only 51% between experienced radiologists [14].

Based on the study by Polanec et al. [9], the primary aim of this study was to correlate the ADC values of potentially malignant prostate lesions with their ISUP grades (International Society of Urological Pathology, [17,18,19]) and thus their clinical significance. To further strengthen the significance of this study, we expanded the inclusion criteria to all histologically confirmed prostate carcinomas regardless of their PI-RADS classification. Furthermore, we added a similar number of histologically confirmed prostate lesions without evidence of carcinoma which were classified as PI-RADS ≥ 3 via mpMRI, summarized as benign prostatic pseudolesions (noPCa or ISUP 0). To avoid unnecessary biopsies in the future, threshold values for the clinical significance of the prostate lesions were determined in a second step, which thus led to the best possible differentiation between csPCa, clinically insignificant prostate cancer (nsPCa) and noPCa. On the one hand, this was determined on the basis of their ADC values alone. On the other hand, because several studies demonstrated a significant correlation between prostate-specific antigen density (PSAD) and csPCa [20,21,22,23,24,25], these two biomarkers were also used together in the form of a ratio: the ADC/PSAD ratio.

## 2. Materials and Methods

This was a retrospective, unicentric study approved by the cantonal ethics committee and conducted at the Winterthur Cantonal Hospital (BASEC Number 2016-01098). Between September 2015 and December 2018, 403 patients with a total of 468 prostate lesions were enrolled, meeting the following inclusion criteria:At least one prostate lesion, assessed according to the current PI-RADS classification at the time of diagnosis, which was histopathologically confirmed in a subsequent MR-TRUS-fused biopsy.

The MR-TRUS-fused biopsy was always targeted and systematic, the latter according to the EAU guidelines (European Association of Urology, [26]), which seems helpful for detecting PCa in areas without obvious lesions [27].

In total, 46 patients (50 prostate lesions) were excluded from the study after reviewing the data. Of these, 7 patients (8 prostate lesions) were excluded due to a non-standardized protocol. This was due to technical problems with the administration of contrast medium, the ADC evaluation or a lack of required sequences, partly because the examination was terminated at the patient’s request. A further 8 patients (9 prostate lesions) were excluded due to the lack of use of an endo-rectal coil and 31 patients (33 prostate lesions) were excluded due to retrospectively not-clearly traceable lesions.

Therefore, 357 patients with a total of 418 prostate lesions remained for the final evaluation.

All imaging was performed on a 3.0 Tesla magnet Philips Achieva (Philips AG Healthcare, Horgen, Switzerland). Prior to the examination, 20 mg butylscopolaminium bromide (Buscopan^®^, Sanofi-Aventis AG, Vernier, Switzerland) was administered intramuscularly and 1 mg glucagon (GlucaGen^®^, Novo Nordisk Pharma AG, Zurich, Switzerland) intravenously. For image acquisition, an endorectal coil (Medrad eCoil BPX30, Bayer Medical Care Inc., Zurich, Switzerland) was used. The image acquisition protocol was standardized and is in accordance with the later-published recommendations of Franiel et al. ([28], Table 1). The ADC maps were automatically calculated from the DWI sequences with different b-factors. The contrast sequences were acquired after a test bolus (NaCl 0.9%) with 20 mL Gadoteridol (ProHance^®^, Bracco Suisse SA, Cadempino, Switzerland; 0.5 mmol/mL, flow 3 mL/s).

The T2w axial, diffusion-weighted sequences (sDWI_b1500) and ADC map were imported into the IntelliSpace Portal (ISP; Philips AG Healthcare, Horgen, Switzerland), an integrated platform with complex image analysis capabilities. The prostate lesions described in the mpMRI report were circumscribed on the ADC map in the ISP using a freehand-contour region of interest (ROI). An average ADC (ADCmean) and a minimal ADC (ADCmin) were measured (Figure 1).

Since the possible avoidance of biopsies was higher for the ADCmin than for the ADCmean (16% vs. 12.4%) in the initial statistical analyses, the ADCmin was used to calculate the ADC/PSAD ratio. This was also in accordance with the study of Donati et al., who found that the 10th percentile ADC is the best indicator for a clinically significant PCa [29].

Additionally, the PI-RADS classification given in the mpMRI, the histopathologically confirmed ISUP grade, the prostate volume calculated in each case via MRI, the PSA in [ng/mL] and thus the PSAD in [ng/mL/cm^3^] were compiled.

The determination of the ADC values in the ISP as described above was carried out entirely by a radiology resident (N.S.) in her 3rd year of training who had previously been instructed in detail. The initial reporting of the prostate mpMRI between September 2015 and December 2018 was performed by specialist radiologists at the Winterthur Cantonal Hospital, according to the then-current PI-RADS v2 classification [12].

The histopathological ISUP grade served as the reference standard for categorial mpMRI findings: ISUP ≥ 2 = csPCa; ISUP 1 = nsPCa; ISUP 0 = noPCa. Here, we are taking the liberty of introducing an ISUP 0 grade for benign lesions, which is helpful for our statistical evaluation.

The statistical software R (version 4.0.2, R Core Team (2021)), with RStudio as the development environment (version 1.3.1073, RStudio PBC), was used. *p*-values ≤ 0.05 were considered statistically significant. Numerical, discrete data, such as ISUP grades, are reported in numbers and percentages.

The ADCmean and ADCmin values, respectively, comprising the ADC/PSAD ratio, were first tested for the possible presence of a normal distribution. As there was no normal distribution, these numerical, continuous values were assessed by means of the Wilcoxon rank sum test when comparing two variables and by means of the Kruskal–Wallis test when comparing more than two variables.

The diagnostic performance of the ADC measurements was evaluated and compared using receiver operating characteristic curves (ROC curves). Using the Youden Index, a measure of the ability of a diagnostic test to balance sensitivity and specificity, optimal threshold calculations for the ADCmean, the ADCmin and the ADC/PSAD ratio were determined. In line with the secondary study objective of avoiding unnecessary biopsies, an exclusion threshold was calculated for each value (sensitivity ≥ 95%) [9].

## 3. Results

In the 357 patients with a total of 418 prostate lesions, 244 (58.4%) histologically confirmed PCa and 174 (41.6%) histologically confirmed noPCa (ISUP 0) were evaluated, the noPCa having been classified as PI-RADS ≥ 3 on initial mpMRI. Of the 244 histologically confirmed PCa lesions, 143 (58.6%) were csPCa (ISUP ≥ 2) and 101 (41.4%) were nsPCa (ISUP 1).

Neither the ADCmean, the ADCmin nor the ADC/PSAD ratio were normally distributed (Shapiro–Wilk test *p* < 0.001).

### 3.1. ADC Values

The performed statistical analyses of the ADC values compared with the histologically confirmed ISUP grades showed a significant negative correlation for both ADCmean values and ADCmin values (Kruskal–Wallis test ADCmean *p* < 0.001; ADCmin *p* < 0.001).

There was a significant difference in the comparison of the ADCmean and ADCmin values of csPCa to the pooled group of nsPCa and noPCa (ns_noPCa), and thus a significant difference in clinical significance (Wilcoxon rank sum test ADCmean *p* < 0.001; ADCmin *p* < 0.001). No significant difference in the ADCmean and ADCmin values was seen when comparing nsPCa and noPCa (Wilcoxon rank sum test ADCmean *p* = 0.27; ADCmin *p* = 0.86), see Figure 2.

In the ROC analysis, the area under the curve (AUC) was 0.69 for the ADCmean and 0.68 for the ADCmin, indicating a moderate discriminatory power of the ADC values for the detection of csPCa. In this study, there was no significant difference between the discriminatory power of the ADCmean and ADCmin.

The optimal threshold value for the diffusion coefficient for the detection of csPCa calculated according to the Youden index was 1239 µm^2^/s for the ADCmean measurements with a corresponding sensitivity of 79% and a specificity of 50.6%. For ADCmin, the threshold was 849 µm^2^/s with a corresponding sensitivity of 76.9% and a specificity of 53.8%.

Assuming that lesions with an average diffusion restriction (ADCmean) greater than 1853 µm^2^/s would have been interpreted as benign (exclusion threshold, sensitivity ≥ 95%), 34 biopsies (34/275, 12.4%) could have been avoided, at the cost of 7 false-negative findings (7/143, 4.9%). If ADCmin with a threshold of 1282 µm^2^/s were used (exclusion threshold, sensitivity ≥ 95%), 44 biopsies could have been avoided (44/275, 16%), at a cost of 7 false-negative findings (7/143, 4.9%). Of these, the false-negative findings in the case of the exclusion threshold based on the ADCmean were four ISUP 2 lesions, two ISUP 3 lesions and one ISUP 4 lesion. When using the exclusion threshold based on the ADCmin, there were three ISUP 2 lesions, three ISUP 3 lesions and one ISUP 4 lesion. Comparing the findings for each threshold, one ISUP 2 lesion, two ISUP 3 lesions and the ISUP 4 lesion were identical.

### 3.2. ADC/PSAD Ratio

The performed statistical evaluation of the ADC/PSAD ratio compared to the histologically confirmed ISUP grade showed a significant negative correlation (Kruskal–Wallis test *p* < 0.001).

There was a significant difference when comparing the ADC/PSAD ratio of csPCa to the ns_noPCa group (Wilcoxon rank sum test *p* < 0.001). Comparison of the ADC/PSAD ratios between nsPCa and noPCa also revealed a significant difference (Wilcoxon rank sum test *p* = 0.0051); see Figure 3.

In the ROC analysis, the AUC was 0.78 for the ADC/PSAD ratio, indicating a moderate-to-good discriminatory power of the ADC/PSAD ratio for the detection of csPCa.

The optimal threshold calculated according to the Youden index is 4904 (µm^2^ × mL × cm^3^)/(s × ng) with a corresponding sensitivity of 74.8% and specificity of 68.4%.

Assuming that lesions with an ADC/PSAD ratio greater than 10,680 (µm^2^ × mL × cm^3^)/(s × ng) would have been interpreted as benign (exclusion threshold, sensitivity ≥ 95%), 87 biopsies (87/275, 31.6%) could have been avoided at the cost of 7 false-negative findings (7/143, 4.9%). Here, the false-negative findings were five ISUP 2 lesions and two ISUP 3 lesions. Both ISUP 3 lesions were also missed by the exclusion thresholds above. For the ISUP 2 lesions, two were missed by the ADCmean threshold and one was missed by the ADCmean and ADCmin thresholds as well. Table 2 summarizes the thresholds calculated above. All ROC curves are summarized in Figure 4.

## 4. Discussion

In this study, we demonstrated a negative correlation between ADC value and ISUP grade and thus the clinical significance of this biomarker. Moreover, this study demonstrated the potential of quantitative ADC evaluation to help avoid unnecessary biopsies of nsPCa or noPCa. On the one hand, avoiding biopsies of nsPCa or noPCa is of economic benefit. The direct costs of MR-TRUS-fused prostate biopsy are further potentiated by its limited availability and the specific equipment and specially trained personnel required [9]. On the other hand, the risks and discomfort for the patient can be avoided. According to Egbers et al., 40% of patients described pain, with a post-interventional hematuria rate of 51% [8]. In addition, approximately 1% of biopsies lead to complications that require or prolong hospitalization [30,31], which in turn also increase costs and thus reduce cost-effectiveness. With the additional use of an ADCmin threshold of 1282 µm^2^/s, 16% of prostate biopsies (44/275) could have been avoided within the study period, with a sensitivity of ≥ 95% [29,32].

Through the correlation of both PSAD and restricted diffusion with clinical significance of prostate lesions, as demonstrated in multiple studies, both values were additionally combined as a ratio in this study [9,11,12,13,14,15,16,20,21,22,23]. Compared to the use of an ADCmin threshold alone, the ADC/PSAD ratio in the ROC analysis not only showed a higher AUC but could have also resulted in the avoidance of nearly twice as many biopsies (31.6%, 87/275) with an applied threshold of 10,680 (µm^2^ × mL × cm^3^)/(s × ng). A sensitivity of ≥95% was still present and the number of false-negative findings remained constant.

As mentioned earlier, this study was guided by the study of Polanec et al. [9]. However, whereas Polanec et al. only evaluated histologically confirmed PCa with a PI-RADS 4 or 5 classification, this study included all histologically confirmed PCa regardless of their PI-RADS classification, as well as a roughly equal number of histologically confirmed noPCa with PI-RADS ≥ 3, and retrospectively evaluated them with respect to average and minimum ADC values [9]. In addition, the study population in this work was over four times larger. Both points strengthen the validity of the results.

Like many studies before, this study confirms the significant negative correlation between ADC value and histopathological outcome [9,11,12,13,14,15,16]. However, only few studies report a quantitative threshold for the diffusion coefficient. Weinreb et al. reported an ADC threshold of 750–900 µm^2^/s for quantitative differentiation between malignant and benign prostate tissue [12]. Polanec et al. also reported a lower ADC threshold of 972 µm^2^/s for quantitative discrimination compared to our study [9]. Other studies have reported a wide range of ADC thresholds, from 930 to 1580 µm^2^/s for malignant and 1610 to 2610 µm^2^/s for normal peripheral prostate tissue [13]. This wide variation is due, in part, to technical factors in diffusion imaging. ADC values depend on many factors, such as the field strength of the static field, the amplitude, duration and interval time between diffusion gradients, and accordingly different b-values. Thus, different quantitative ADC limits can be at least partially explained [13]. Due to the heterogeneity of hardware, image acquisition and post-processing, the definition of a generally valid threshold should therefore be interpreted with caution and should ideally be determined on an institutional basis [33,34].

An applied ADCmin threshold of 1282 µm^2^/s resulted in a potential avoidance of 16% of biopsies in this study. Although the area under the curve was minimally lower for the ADCmin (AUC = 0.68) than for the ADCmean (AUC = 0.69), the potential biopsy avoidance rate was higher for the ADCmin than for the ADCmean (16% vs. 12.4%).

The lower avoidance rate of 16%, compared with the 33% of Polanec et al., may be explained by the different study populations [9]. As mentioned above, not only was the population of this study over four times larger, but this study also included all PCa regardless of their PI-RADS classification, as well as noPCa with PI-RADS ≥ 3. In contrast, Polanec et al. and Gaur et al. evaluated only PI-RADS 4 and 5 lesions, and thus obtained an AUC of up to 0.87 in their studies [9,16]. Therefore, in addition to the above-mentioned technical factors, the somewhat-lower thresholds of the other studies compared to this study may also be explained, since patients with noPCa misread as PI-RADS ≥ 3 were not evaluated in the comparative studies.

In their study, Westphalen et al. demonstrated the improvement of characterization of PI-RADS 4 and 5 lesions by independently including restricted diffusion as well as PSAD [22]. As mentioned above, this and multiple other studies demonstrated the negative correlation between ADC value and histologic grade as well as the correlation between PSAD and histologic grade [9,11,12,13,14,15,16,20,21,22,23]. Due to this, the two values were combined in this study to potentially potentiate the significance.

Quantitative evaluation of the ADC/PSAD ratio has not yet been carried out in any other study, as far as could be researched at the time of this study.

With an almost-doubled avoidance rate of biopsies (31.6% vs. 16%) compared to the sole use of an ADCmin threshold, the ADC/PSAD ratio represents a promising biomarker, especially since the number of false negative findings was kept constant (7/143, 4.9%). The AUC of 0.78 is also higher than the AUC of the ADC limits (0.69 for the ADCmean and 0.68 for the ADCmin), showing even better discriminatory power. Lee et al. found a similar, even slightly better AUC of 0.84 while quantitative evaluating PSAD together with the already mentioned 10th percentile of ADC of Donati et al., which they named “model 4” out of six models and which was the best of the quantitative models [29,32].

In addition, the ADC/PSAD ratio also exhibits significant discrimination between nsPCa and noPCa, which was not shown when using ADC values alone (*p* = 0.0051 vs. *p* = 0.27 for ADCmean and *p* = 0.86 for ADCmin). Thus, the ADC/PSAD ratio holds promise not only for the detection of csPCa but also for the differentiation between nsPCa and noPCa. In the case of nsPCa, follow-up (active surveillance) without prior biopsy could then be considered. Further studies need to confirm the superiority of this ratio.

This study was limited by its retrospective and unicentric design. However, in return, the unicentric design also guaranteed consistency in hardware, image acquisition and image analysis. The measurement of ADC values for the quantitative analysis by only one person is on the one hand a limitation, due to possible bias, but on the other hand also guarantees a certain consistency of analysis. The large study population compared to other studies of this kind increases the significance of this study. In addition, all histologically confirmed carcinomas were evaluated irrespective of their PI-RADS classification and an almost equally large group of noPCa was added.

There is some selection bias in the noPCa, as only PI-RADS ≥ 3 lesions were evaluated here. The last limitation of this study is the reference standard. According to other studies, the Gleason score of biopsies, and hence the ISUP grade, differs slightly compared to the Gleason score of prostatectomies due to sampling errors as well as the multifocality and heterogeneity of PCa [13].

## 5. Conclusions

In summary, quantitative analysis of the diffusion coefficients of prostate lesions confirms a negative correlation with ISUP grades and thus with clinical significance. Furthermore, the quantitative analysis, considering the evaluated threshold values, for ADCmin and the ADC/PSAD ratio shows a great potential for avoiding prostate biopsies in up to 31.6% of lesions. In addition, the ADC/PSAD ratio promises the possibility of distinguishing between nsPCa and noPCa, and thus offers the alternative of active surveillance without prior biopsy. In the absence of comparative studies regarding the ADC/PSAD ratio, further studies are needed to confirm its superiority.

## Figures and Tables

**Figure 1 cancers-15-05183-f001:**
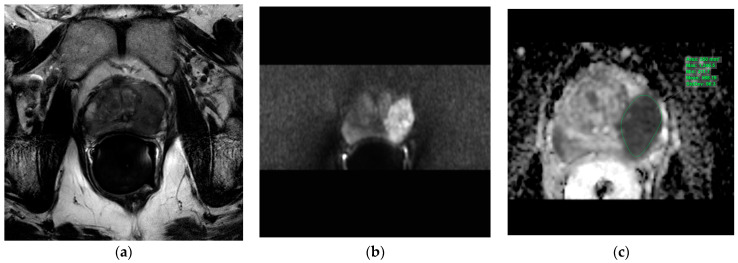
Evaluation of the apparent diffusion coefficients in the IntelliSpace Portal (ISP; Philips AG Healthcare, Horgen, Switzerland). Illustration of the three transversal MRI sequences loaded into the IntelliSpace Portal system for apparent diffusion coefficient evaluation. Green indicates a freehand-contour ROI around a suspicious lesion in the peripheral zone on the left. (**a**): T2w, (**b**): DWI b1500, (**c**): ADC map. ADC = apparent diffusion coefficient, DWI = diffusion-weighted imaging, MRI = magnetic resonance imaging, ROI = region of interest, T2w = T2-weighted.

**Figure 2 cancers-15-05183-f002:**
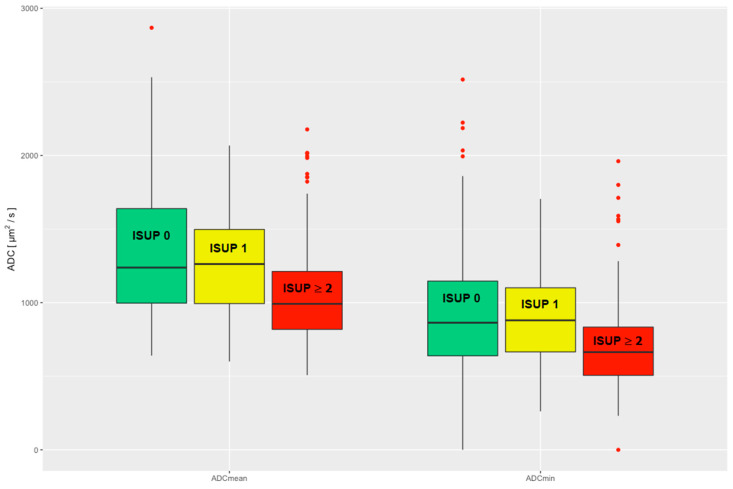
Evaluated ADCmean and ADCmin are divided into three clinical groups: noPCa, ISUP 0, green; nsPCa, ISUP 1, yellow; csPCa, ISUP ≥ 2, red. ADC = apparent diffusion coefficient, ISUP = International Society of Urological Pathology, noPCa = benign lesions/pseudolesions, nsPCa = clinically insignificant prostate cancer, csPCa = clinically significant prostate cancer.

**Figure 3 cancers-15-05183-f003:**
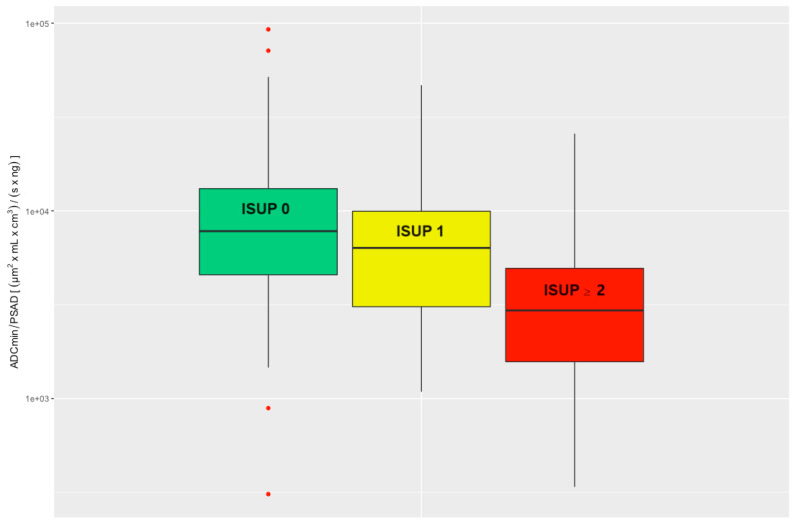
Evaluated ADC/PSAD ratio divided into three clinical groups: noPCa, ISUP 0, green; nsPCa, ISUP 1, yellow; csPCa, ISUP ≥ 2, red. ADC = apparent diffusion coefficient, ISUP = International Society of Urological Pathology, PSAD = prostate-specific antigen density, noPCa = benign lesions/pseudolesions, nsPCa = clinically insignificant prostate cancer, csPCa = clinically significant prostate cancer.

**Figure 4 cancers-15-05183-f004:**
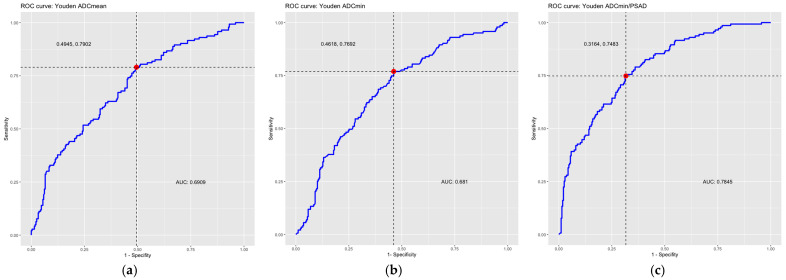
ROC curves including optimal thresholds (red dots, dotted lines) according to Youden Index and AUC. ROC curve of (**a**) the average ADC values (ADCmean), (**b**) the minimum ADC values (ADCmin), (**c**) the ADC/PSAD ratio. The area under the curve (AUC) and the optimal threshold values calculated according to the Youden Index are also given. ADC = apparent diffusion coefficient, AUC = area under the curve, PSAD = prostate-specific antigen density, ROC = receiver operating characteristic.

**Table 1 cancers-15-05183-t001:** Acquired mpMRI prostate protocol.

Sequence Weighting	T2	T2	T2	DWI	T1	T1	T1
sequence type	SE	SE	SE	EPI	SE	THRIVE	Dixon
orientation	sag	tra	cor	tra	tra	tra	tra
TR [ms]	3200	4300	4200	4200	650	3.1	4600
TE [ms]	130	130	120	70	7.8	1.5	0
matrix [pixel^2^]	880^2^	560^2^	720^2^	192^2^	560^2^	448^2^	224^2^
resolution [mm^2^]	0.25^2^	0.2857^2^	0.25^2^	0.9375^2^	0.2857^2^	1.1931^2^	0.9375^2^
FOV [mm^2^]	220^2^	160^2^	180^2^	180^2^	160^2^	267^2^	420^2^
thickness [mm]	3	3	3	3	3	3	3
flip angle [°]	90	90	90	90	90	10	10
b-value [s/mm^2^]				100, 950, 1500			
temporal resolution [s]						5	
spacing [mm]	3	3	3	3	3	1.4	1.51
acceleration factor	1.5	1.3	1.7		1.5		
average	1	1	1	1	2	1	1

Summary of all MRI sequences of the prostate protocol, including specific parameters. T2 = T2-weighted, DWI = diffusion-weighted imaging, T1 = T1-weighted, SE = spin-echo, EPI = echo-planar imaging, THRIVE = T1-weighted high-resolution isotropic volume examination dynamic sequence post contrast, Dixon = T1-weighted Dixon sequence post contrast, sag = sagittal, tra = transversal, cor = coronal, TR = repetition time, TE = echo time, FOV = field of view.

**Table 2 cancers-15-05183-t002:** Threshold values.

	Method	Threshold [*]	Sensitivity [%]	Specificity [%]	TP	FN	FP	TN
**ADCmean**	Youden	1239	79.0	50.6	113	30	136	139
	Exclusion	1853	95.1	12.4	136	7	241	34
**ADCmin**	Youden	846	76.9	53.8	110	33	127	148
	Exclusion	1282	95.1	16.0	136	7	231	44
**ADC/PSAD**	Youden	4904	74.8	68.4	107	36	87	188
	Exclusion	10680	95.1	31.6	136	7	188	87

Different threshold values according to Youden Index and exclusion threshold (criterion sensitivity ≥ 95%) calculated with corresponding sensitivities, specificities and TP, FN, FP and TN. The ADC/PSAD ratio was calculated with the ADCmin values. ADC = apparent diffusion coefficient, FN = false negative, FP = false positive, TN = true negative, TP = true positive, ***** units for ADC are [µm^2^/s], for ADC/PSAD [(µm^2^ × ml × cm^3^)/(s × ng)].

## Data Availability

The datasets used and/or analyzed during the current study are available from the corresponding author on reasonable request.

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
