# Peer review of "Quantitative Evaluation of Apparent Diffusion Coefficient Values, ISUP Grades and Prostate-Specific Antigen Density Values of Potentially Malignant PI-RADS Lesions"

_cancers, 2023, doi:10.3390/cancers15215183_

Round 1

Reviewer 1 Report

Comments and Suggestions for Authors

I think this is an excellent contribution and highly important to all physicians involved with PCa patients.  I converted the manuscript from PDF to Word so I could more easily use the Review tool in Word to offer my recommendations to you.  Your grammar is overall excellent and I only found a few areas where I think the content could be presented more simply and clearly to the reader. 

I also feel that you could add text to some of your figures so that they could stand alone as teaching vehicles.  

This was the most enjoyable review I have done in the last two years. I hope the other reviewers concur with my assessment. 

Congrats to all of you involved. 

Comments on the Quality of English Language

See above. I found the manuscript to be very well written.  There are only a few sentences that could be restructured for clarity.  

Reviewer 2 Report

Comments and Suggestions for Authors

Round 2

Reviewer 2 Report

Comments and Suggestions for Authors

Thanks for addressing my comments.